# Simultaneous evaluation of metabolomic and inflammatory biomarkers in children with different body mass index (BMI) and waist-to-height ratio (WHtR)

Erika Chavira-Suárez[1], Cecilia Rosel-Pech[1,2], Ernestina Polo-Oteyza[3], Mónica Ancira-Moreno[1,4], Isabel Ibarra-González[5], Marcela Vela-Amieva[6], Noemi Meraz-Cruz[1], Carlos Aguilar-Salinas[7], Felipe Vadillo-Ortega[1,8]*

1 Unidad de Vinculación Científica de la Facultad de Medicina, Universidad Nacional Autónoma de México en el Instituto Nacional de Medicina Genómica, Mexico City, Mexico, 2 Unidad de Investigación Biomédica en Inmunología e Infectología, Hospital de Infectología "Dr. Daniel Méndez Hernández" Centro Médico Nacional La Raza, Instituto Mexicano del Seguro Social, Mexico City, Mexico, 3 Fondo Nestlé para la Nutrición, Fundación Mexicana Para la Salud, Mexico City, Mexico, 4 Health Department, Universidad Iberoamericana, A.C., Mexico City, Mexico, 5 Instituto de Investigaciones Biomédicas, Universidad Nacional Autónoma de México, Mexico City, Mexico, 6 Laboratorio de Errores Innatos del Metabolismo y Tamiz, Instituto Nacional de Pediatría, Secretaría de Salud, Mexico City, Mexico, 7 Dirección de Nutrición, Instituto Nacional de Ciencias Médicas y Nutrición Salvador Zubirán, Mexico City, Mexico, 8 Dirección de Investigación, Instituto Nacional de Medicina Genómica, Mexico City, Mexico

☯ These authors contributed equally to this work.
* fvadillo@inmegen.gob.mx

**Data Availability Statement:** All relevant data are within the manuscript.

## Abstract

Metabolic disturbances and systemic pro-inflammatory changes have been reported in children with obesity. However, it is unclear the time-sequence of metabolic or inflammatory modifications during children obesity evolution. Our study aimed to quantify simultaneously metabolomic and inflammatory biomarkers in serum from children with different levels of adiposity. For this purpose, a cross-sectional study was used to perform targeted metabolomics and inflammatory cytokines measurements. Serum samples from children between six to ten years old were analyzed using either body mass index (BMI) or waist-to-height ratio (WHtR) classifications. One hundred and sixty-eight school-aged children were included. BMI classification in children with overweight or obesity showed altered concentrations of glucose and amino acids (glycine and tyrosine). Children classified by WHtR exhibited imbalances in amino acids (glycine, valine, and tyrosine) and lipids (triacyl glycerides and low-density lipoprotein) compared to control group. No differences in systemic inflammation biomarkers or in the prevalence of other results were found in these children. Abnormal arterial blood pressure was found in 32% of children with increased adiposity. In conclusion, obesity in school-aged children is characterized by significant metabolic modifications that are not accompanied by major disturbances in circulating concentrations of inflammatory biomarkers.

**Funding:** Field work of this study was supported from Nestlé, S.A de C.V. under agreement 33059-2283-25-X-12 with Universidad Nacional Autónoma de México. I would like to state that Ernestina Polo-Oteyza, M.Sc. contributed as an author to this paper. She has a commercial affiliation as Director of Fondo Nestlé para la Nutrición in the Fundación Mexicana para la Salud. The funder provided support in the form of salaries for author EPO, but did not have any additional role in the study design, data collection and analysis, decision to publish, or preparation of the manuscript. The specific roles of this author are articulated in the 'author contributions' section.

## Introduction

An increasing and alarming prevalence of overweight (OV) and obesity (OB) in children has been documented worldwide. Chile, USA, New Zealand and Mexico are among the countries with the highest reported prevalence [1–3]. Determinants of pediatric obesity are a complex interplay between high energy intake and a low energy expenditure with genetic and epigenetic factors, neuroendocrine modulation, intrauterine exposures, sleeping habits, physical activity, and socioeconomic level [4]. Abnormal adiposity starting in childhood introduces a higher risk for early development of metabolic syndrome, type 2 diabetes and cardiovascular diseases, among many other complications [5,6]. Current understanding of the deleterious effects of increased adiposity points to lipotoxicity as the initial mechanism of disease.

Lipotoxicity is a term reflecting the damaging effects of systemic chronic increased availability of free fatty acids (FFA) that induces reactive oxygen species (ROS) production, impaired mitochondrial function and metabolic adjustments in several tissues, accompanying systemic insulin resistance [7] and release of pro-inflammatory cytokines [8]. Metabolic adjustments include accumulation of FFA in non-adipose tissues such as skeletal muscle, heart, pancreas, liver and kidney, and the wide use of lipids as a major source for catabolism, associated to incomplete fatty acids β-oxidation [9] and suppression of the insulin intracellular pathways that mediate glucose uptake in tissues [10]. Also, FFA accumulation can increase the formation of abnormal autophagosomes in beta cells, inducing endoplasmic reticulum stress which it could be a factor in the progression from obesity to diabetes [11].

Increased availability of circulating FFA can exceed the adipose tissue storage capacity and directly stimulate toll-like receptor 4 (TLR4) in blood mononuclear cells, inducing the secretion of pro-inflammatory cytokines [8]. In addition, secretions of mediators such as retinol-binding protein 4 (RBP-4), leptin, adipsin, vaspin, resistin, c-reactive protein (CRP), adiponectin, and omentin-1 promote imbalance in glucose and lipid metabolism, breaking down the insulin-sensitizing mechanisms, signal transduction, and triggering insulin resistance, ROS overproduction, energy uncoupling in mitochondria, cellular apoptosis and necrosis and reduced nitric oxide (NO) production [12,13].

Changes in metabolic substrates availability by physiological and pathophysiological conditions could be measured by diverse biochemical and molecular technical approaches. Currently, metabolomics is becoming a powerful tool for identifying metabolic signatures associated with healthy or unhealthy phenotypes. For example, the metabolomic analysis in blood samples from children with obesity and type 2 diabetes enables to detect metabolites changes that have pathophysiological relevance in clinical practice for hyperinsulinism and insulin resistance treatment [14]. The screening for metabolic profiling in children with different obesogenic background may facilitate individual or personalized pediatric interventions [15]. However, it still unclear if alterations in metabolic substrates are a consequence of inflammatory factors that responded to obesity condition or that metabolic imbalances promote pro-inflammatory response in children.

Few efforts have been directed to evaluate the specific contribution, timing and interactions of the metabolic and inflammatory mechanisms of damage in obesity, especially in children in which the consequences of these disturbances can initiate an early-life process of accumulating damaging effects on health. In this work we compare the presence of specific metabolomic disturbances and levels of blood pro-inflammatory mediators in a cross-sectional study of school-aged children stratified both by body mass index (BMI) and waist-to-height ratio (WHtR).

## Methods

### Subjects

One hundred and sixty-eight Mexican children between six to ten years old were invited to participate through a signed consent form by them and their parents. They were randomly selected from a cohort of 1,312 school-aged children attending 2012–2013's cycle of public elementary school at the City of Toluca, State of Mexico, Mexico. They were classified by their BMI and WHtR for further metabolomic and immunological analyses. Children were excluded if previous diagnosis of type 1 and type 2 diabetes, dyslipidemia, hypertension or inflammatory diseases was presented. Sample size was calculated according to a previously published study in school-aged Mexican children for finding differences among mean values of serum cytokine concentrations between BMI groups [5].

Institutional Review Board of the Faculty of Medicine, National Autonomous University of Mexico (UNAM) granted authorization for this study (Register number: 2013–89) and, it has been conducted according to the principles expressed in the Declaration of Helsinki.

### Anthropometric measurements

Height was measured with a stadiometer 217 model (Seca, Germany). Weight was obtained with a flat scale 876 model (Seca, Germany), and circumference of waist was measured with a measuring tape (Seca, Germany). Anthropometric measurements were assessed by standardized personnel, using Lohman techniques [16]. Arterial blood pressure, pulse and respiration rate were measured at rest. Blood pressure was obtained according to American Heart Association [17]. The BMI was calculated and BMI Z-scored according with World Health Organization (WHO) reference values [18]. The WHtR was calculated as the waist circumference (cm) divided by the height (cm). LMS tables for calculating WHtR in Z-scores and centiles based on National Health and Nutrition Examination Survey (NHANES) were applied to make comparisons [19]. Systolic (SBP) and diastolic (DBP) blood pressures (mmHg) percentiles for gender, age, and height were calculated following Eunice Kennedy Shriver National Institute (NICHD) recommendations.[20] Children with OV and OB were classified by BMI, using Z-score > 2.0 to ≤ 3.0 and > 3.0, respectively.[18] Cardiovascular risk group (CVR) in children was defined by WHtR at 65[th] percentile in girls and 77[th] percentile in boys [19].

### Metabolomics

Venous blood samples were obtained under fasting conditions, and serum separated no later than fifteen minutes after venipuncture. Glucose (Gluc), triacylglycerols (TAG), total cholesterol (TC), high density lipoprotein cholesterol (HDL-c), and low-density lipoprotein cholesterol (LDL-c) were measured using a clinical chemistry analyzer (ISE SRL, Miura 200). Eleven L-amino acids including arginine (Arg), citruline (Cit), glycine (Gly), alanine (Ala), leucine (Leu), L-methionine (Met), L-phenylalanine (Phe), tyrosine (Tyr), Valine (Val), Ornithine (Orn), Proline (Pro), and twelve acylcarnitines including carnitine (C0), acetylcarnitine (C2), propionylcarnitine (C3), isobutyril-L-carnitine (C4), isovalerylcarnitine (C5), hexanoylcarnitine (C6), octanoylcarnitine (C8), decanoylcarnitine (C10), dodecanoylcarnitine (C12), tetradecanoylcarnitine (C14), L-palmitoylcarnitine (C16), and sterearoylcarnitine (C18), were measured by tandem mass spectrometry (Micromass Quattro micro™ API, Perkin Elmer) using the NeoLinx 4.1 software (Perkin Elmer). For each subject serum insulin (Ins) level was measured by ultrasensitive immunoassay method in a Beckman Coulter Access 2 system (Beckman Coulter Ireland, Inc.). From the Gluc and Ins determinations, HOMA-IR was calculated as follows: fasting blood Gluc (mmol/L) × fasting Ins (microIU/mL)/22.5.

## Cytokine measurements

Fourteen cytokines including interleukins IL-1α, IL-1β, IL-2, IL-6, IL-10, IL-17, IL-1RA, IL-12p40, sIL-2RA, interferon gamma-induced protein 10 (IP10), tumor necrosis factor-α (TNFα), vascular endothelial growth factor (VEGF), macrophage inflammatory proteins (MIP1α), and MIP1β were measured in serum using Milliplex Multiplex Assays (Millipore, Burlington, MA, USA). Media fluorescence intensity, calculated from duplicates from each sample, was analyzed using the Luminex-100 software system version 1.7 (Luminex). Observations under lower limit of detection (LOD) for each cytokine were transformed for the analysis using LOD/2 square root method [21].

## Statistical analysis

Numerical data are reported as the mean ± SD and its corresponding proportion by each group of BMI and WHtR classifications. Comparisons between group means were computed using 2-way ANOVA with post Fisher's LSD test, Kruskal Wallis with Dunnett's multiple comparison test, and U Mann Whitney where appropriate. Metabolites and cytokines were plotted by partial least squares-discriminant analysis (PLS-DA) and compared by multiple logistic regressions using the covariables of gender and age to distinguish main contributors in each group. Analyses were conducted using SPSS version 19 and MetaboAnalyst 3.0. Statistical significance was set at $p \leq 0.05$.

## Results

Among the 168 children included in this study, the mean age was 8.8 ± 1.3 years and approximately half of them were girls (46%; n = 78) and the other half boys (54%; n = 90). According to BMI classification, 77 (45%) of children corresponded to normal weight (NW), 45 (27%) to OV, and 46 (28%) to OB. Significant increments of DBP and SBP were found in children with OV and OB (Table 1).

Children classified by WHtR showed that 84 (50%) of them belonged to the normal group (NG), and the other 84 (50%) belonged to the cardiovascular risk group (CVR). SBP and DBP values were significantly higher in children classified as CVR than NG (Table 2).

Further analysis of blood pressure values was processed using WHtR percentile ranks according to National Institutes of Health (NIH) and National Heart, Lung, and Blood

**Table 1. Characteristics of 168 schoolchildren classified by BMI.**

| Anthropometric values | Normal weight | | | Overweight | | | Obesity | | |
|---|---|---|---|---|---|---|---|---|---|
| | Girls | Boys | Both | Girls | Boys | Both | Girls | Boys | Both |
| | N (%) | | | N (%) | | | N (%) | | |
| | 36 (21) | 41 (24) | 77 (45) | 26 (16) | 19 (11) | 45 (27) | 16 (10) | 30 (18) | 46 (28) |
| | Mean ± SD | | | Mean ± SD | | | Mean ± SD | | |
| Weight (kg) | 26 ± 5 | 24 ± 4 | 25 ± 5 | 37 ± 7[b] | 33 ± 6[a] | 35 ± 7[a,b] | 44 ± 8[c] | 42 ± 9[c] | 43 ± 9[c] |
| Height (cm) | 130 ± 9 | 127 ± 9 | 128 ± 9 | 136 ± 10[b] | 130 ± 10[a] | 133 ±10[a,b] | 136 ± 8[b] | 135 ± 10[a,b] | 135 ± 9[a,b] |
| BMI (kg/m$^2$) | 15 ± 1 | 15 ± 1 | 15 ± 1 | 20 ± 2[a] | 19 ± 1[a] | 19 ± 2[a] | 24 ± 2[b] | 23 ± 2.5[b] | 23 ± 2[b] |
| SBP (mmHg) | 103 ± 15[a] | 104 ± 11[a] | 103 ± 13[a] | 106 ± 1[a,b] | 111 ± 9[b] | 108 ±12[b] | 107 ± 9[a,b] | 109 ± 14[a,b] | 109 ±13[b] |
| DBP (mmHg) | 63 ± 11[a] | 59 ± 11[a] | 61 ± 11[a] | 64 ± 13[a,b] | 66 ± 11[b] | 65 ± 12[a,b] | 66 ± 10[a,b] | 65 ± 11[b] | 65 ± 10[b] |

Descriptive results are expressed as the means ± SD, indicating the number of subjects (N) and the corresponding percentage (%). Two-way ANOVA post Fisher's LSD test was used; statistical difference (p<0.05) between genders of the same group or between groups is shown with different letters. BMI: body mass index, SBP: systolic blood pressure, DBP: diastolic blood pressure

**Table 2. Characteristics of 168 schoolchildren classified by WHtR.**

| Anthropometric values | Normal | | | Cardiovascular Risk | | |
|---|---|---|---|---|---|---|
| | Girls ($\leq 65^{th}$) | Boys ($\leq 77^{th}$) | Both | Girls ($> 65^{th}$) | Boys ($> 77^{th}$) | Both |
| | N (%) | | | N (%) | | |
| | 35 (21) | 49 (29) | 84 (50) | 43 (26) | 41 (24) | 84 (50) |
| | Mean ± SD | | | Mean ± SD | | |
| Weight (kg) | 27 ± 7 | 25 ± 5 | 26 ± 6 | 39 ± 9[a] | 40 ± 10[a] | 39 ± 9[a] |
| Height (cm) | 131 ± 10[a] | 128 ± 8[a] | 130 ± 9[a] | 135 ± 9[a,b] | 133 ±11[b] | 134 ± 10[b] |
| Waist (cm) | 56 ± 5 | 56 ± 5 | 56 ± 5 | 75 ± 8[a] | 76 ± 9[a] | 75 ± 8[a] |
| WHtR/age | 0.43 ± 0.03 | 0.43 ± 0.03 | 0.43 ± 0.03 | 0.55 ± 0.04[a] | 0.57 ± 0.05[b] | 0.56 ± 0.04[a,b] |
| SBP (mmHg) | 101±14 | 105 ±11 | 103 ± 12 | 108 ± 13[a] | 110 ± 12[a] | 109 ±13[a] |
| DBP (mmHg) | 62 ± 11[a,b] | 60 ± 12[a] | 61 ± 12[a,b] | 66 ± 11[b,c] | 65 ± 10[c] | 65 ±11[c] |

Descriptive results are expressed as the means ± SD, indicating the number of subjects (N) and the corresponding percentage (%). Two-way ANOVA post Fisher's LSD test was used; statistical difference (p<0.05) between genders of the same group or between groups is shown by different letters. WHtR: waist to height ratio, SBP: Systolic blood pressure, DBP: diastolic blood pressure.

Institute (NHLBI)[20]. This kind of analysis allowed us to identify children with different levels of hypertension risk: normal (<$90^{th}$), prehypertension ($\geq 90^{th}$ <$95^{th}$), hypertension state 1 ($\geq 95^{th}$ <$99^{th}$), and hypertension state 2 ($\geq 99^{th}$). Significative difference in prevalence of abnormal SBP was found in children classified as CVR (OR: 2.776, 95% IC: 1.14–6.76; p = 0.011) (Table 3).

In order to clarify if school-aged children with obesity-associated risks could has metabolic substrates altered, we measure metabolites involved in glucose, amino acids, and lipids metabolisms. Our findings showed that metabolomic profiles overlapped between groups either classified by BMI or WHtR (Fig 1).

In bivariate analysis, the values of Gluc, TAG, TC, LDL-c, C0, C14, C16, C3, C5, Gly, Val, and Tyr showed differences between NW and OB; while the values of Ins, HOMA-IR, C0, and C6 showed exclusively differences between NW and OV. The differences observed between NW and OB were pretty similar to those of NG and CVR, but in WHtR classification acylcarnitines C0, C6, C5 and the amino acid Gly expressed no differences between both groups;

**Table 3. Classification by WHtR and blood pressure.**

| Classification | Systolic Blood Pressure (mm Hg) | | | | | |
|---|---|---|---|---|---|---|
| | Normal | | | Cardiovascular Risk | | |
| | Girls | Boys | Both | Girls | Boys | Both |
| | Mean ± SD (N) | | | Mean ± SD (N) | | |
| Normal (< $90^{th}$) | 99 ±12 (32) | 102 ± 8 (44) | 101 ±10 (76) | 103 ±9 (35) | 105 ±10 (30) | 104 ± 10 (65) |
| Prehypertension ($\geq 90^{th}$ < $95^{th}$) | 120 (2) | 119 (2) | 119 ± 2 (4) | 119 ± 1 (3) | 118 ± 2 (4) | 118 ± 2 (7) |
| Hypertension state I ($\geq 95^{th}$ < $99^{th}$) | (0) | (0) | (0) | 124 (2) | 122 ± 2 (5) | 123 ± 2 (7) |
| Hypertension state II ($\geq 99^{th}$) | 129 (1) | 133 ± 10 (3) | 132 ± 8 (4) | 137 ± 2 (3) | 138 (2) | 137 ± 5 (5) |
| | Diastolic Blood Pressure (mm Hg) | | | | | |
| Normal (< $90^{th}$) | 59 ± 7 (30) | 57 ± 7 (45) | 58 ± 7 (75) | 62 ± 9 (36) | 63 ± 9 (36) | 63 ± 9 (72) |
| Prehypertension ($\geq 90^{th}$ < $95^{th}$) | 79 ± 2 (3) | 77 (1) | 79 ± 2 (4) | 80 (2) | 79 (2) | 79 ± 2 (4) |
| Hypertension state I ($\geq 95^{th}$ < $99^{th}$) | 85 (1) | (0) | 85 (1) | 83 ± 1 (4) | 81 (2) | 82 ± 2 (6) |
| Hypertension state II ($\geq 99^{th}$) | 89 (1) | 95 ± 11 (3) | 94 ± 6 (4) | 88 (1) | 83 (1) | 86 ± 4 (2) |

WHtR percentile ranks classification in children according to NHLBI Health Information Center of the National Institutes of Health [20].

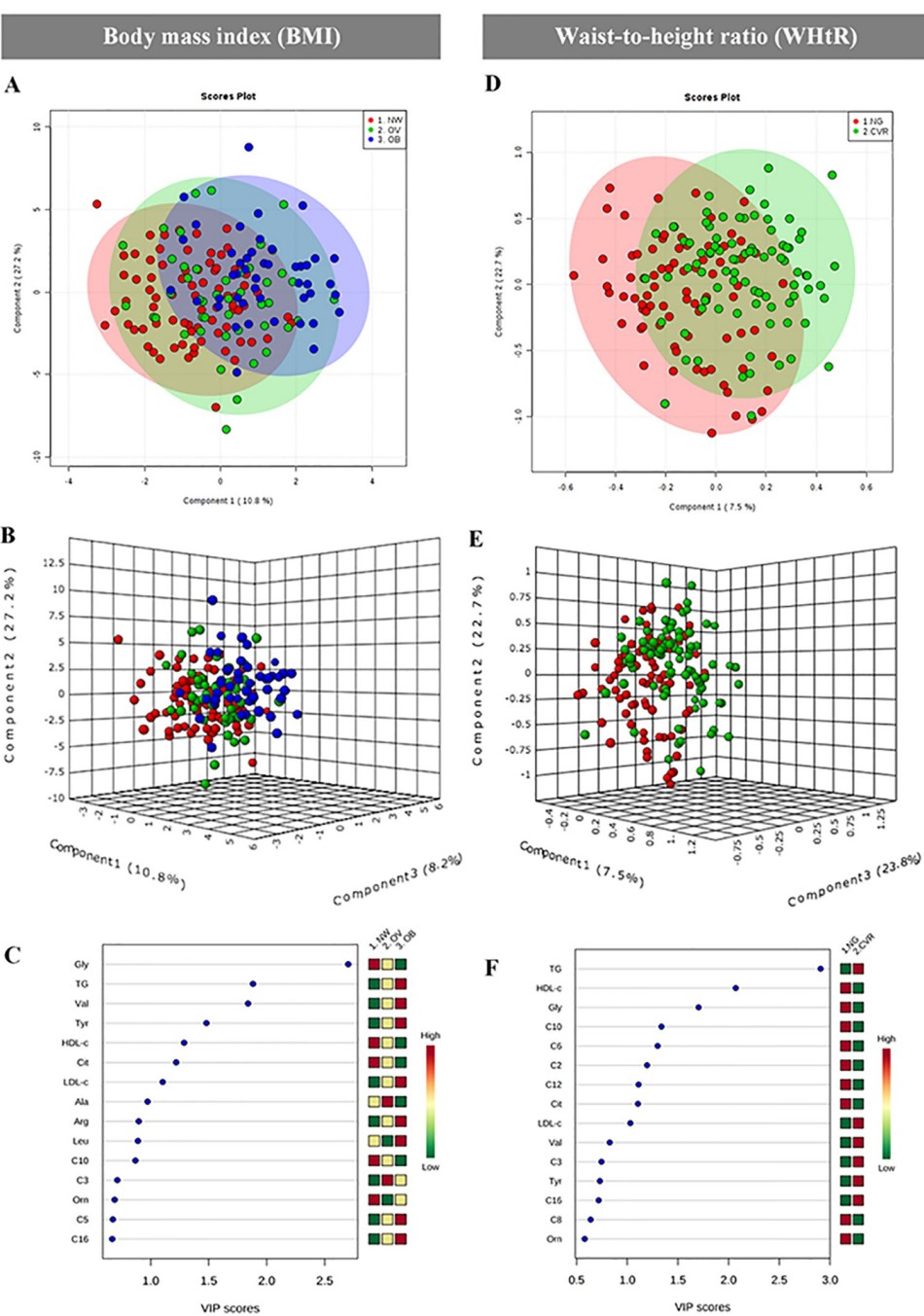

**Fig 1. Metabolites profile in serum from schoolchildren classified by BMI and WHtR.** (A—C) BMI classification; (D—F) WHtR classification. PLS-DA plots and loadings plots showing the maximum covariance between metabolites concentrations and the largest effect by children adiposity group.

instead of it, HDL-c value was lower in CVR than NG (S1 Table). Despite this, after multiple logistic regression analysis only the Gluc, Gly, Phe, and Orn were different in children with OV/OB, as well as LDL-c, Gly, Val, and Orn were different in CVR; age and gender were not associated (Table 4).

**Table 4. Multivariable model for prediction of metabolic alterations in schoolchildren.**

| Variable | BMI classification | | | | WHtR classification | | | |
|---|---|---|---|---|---|---|---|---|
| | β | S. E. | χ2 | Prob> χ2 | β | S. E. | χ2 | Prob> χ2 |
| Intercept | 11.5 | 4.7 | 6.00 | **0.014***  | 8.99 | 4.8 | 3.45 | 0.063 |
| Gender | 0.146 | 0.26 | 0.32 | 0.571 | 0.019 | 0.25 | 0.01 | 0.940 |
| Age | 0.080 | 0.20 | 0.17 | 0.682 | -0.127 | 0.20 | 0.40 | 0.529 |
| Gluc | -0.105 | 0.05 | 4.93 | **0.026***  | -0.059 | 0.04 | 1.77 | 0.183 |
| Ins | -2.06 | 1.9 | 1.13 | 0.287 | -1.52 | 1.9 | 0.64 | 0.425 |
| HOMA-IR | 7.67 | 8.2 | 0.88 | 0.347 | 5.81 | 8.0 | 0.52 | 0.470 |
| TAG | -0.003 | 0.01 | 0.19 | 0.663 | -0.010 | 0.01 | 1.49 | 0.223 |
| TC | -0.002 | 0.02 | 0.02 | 0.902 | 0.014 | 0.02 | 0.60 | 0.439 |
| LDL-c | -0.028 | 0.02 | 1.55 | 0.213 | -0.050 | 0.02 | 4.17 | **0.041***  |
| HDL-c | 0.036 | 0.03 | 2.01 | 0.156 | 0.034 | 0.03 | 1.52 | 0.218 |
| C0 | -0.001 | 0.11 | 0.00 | 0.989 | -0.000 | 0.11 | 0.00 | 0.997 |
| C2 | -0.115 | 0.73 | 0.03 | 0.874 | 0.235 | 0.79 | 0.09 | 0.764 |
| C4 | 13.5 | 13 | 1.03 | 0.310 | 15.3 | 13 | 1.33 | 0.249 |
| C6 | 81.9 | 43 | 3.68 | 0.055 | 86.3 | 45 | 3.73 | 0.054 |
| C8 | 8.71 | 14 | 0.41 | 0.523 | 17.1 | 13 | 1.64 | 0.200 |
| C10 | -11.1 | 19 | 0.33 | 0.567 | -27.6 | 19 | 2.12 | 0.145 |
| C12 | 0.386 | 41 | 0.00 | 0.993 | 44 | 41 | 1.12 | 0.290 |
| C14 | -91.9 | 110 | 0.70 | 0.404 | -221 | 122 | 3.25 | 0.072 |
| C16 | 12.3 | 33 | 0.14 | 0.706 | 18.9 | 33 | 0.32 | 0.571 |
| C18 | 90.4 | 67 | 1.82 | 0.178 | 66.0 | 68 | 0.93 | 0.334 |
| C3 | 12.3 | 13 | 0.87 | 0.350 | 5.1 | 13 | 0.16 | 0.685 |
| C5 | -61.3 | 32 | 3.64 | 0.0565 | -19.0 | 32 | 0.36 | 0.549 |
| Gly | 0.033 | 0.01 | 9.75 | **0.009***  | 0.037 | 0.01 | 11.49 | **0.001***  |
| Ala | -0.005 | 0.01 | 0.22 | 0.642 | 0.007 | 0.01 | 0.31 | 0.575 |
| Met | 0.282 | 0.18 | 2.59 | 0.108 | -0.077 | 0.17 | 0.22 | 0.642 |
| Leu | -0.038 | 0.05 | 0.67 | 0.413 | 0.048 | 0.05 | 1.12 | 0.289 |
| Val | -0.052 | 0.05 | 1.31 | 0.252 | -0.107 | 0.05 | 5.44 | **0.020***  |
| Phe | 0.180 | 0.07 | 5.97 | **0.015***  | 0.054 | 0.07 | 0.58 | 0.447 |
| Tyr | -0.095 | 0.05 | 3.51 | 0.061 | -0.076 | 0.05 | 2.72 | 0.099 |
| Arg | -0.078 | 0.05 | 2.73 | 0.099 | -0.065 | 0.04 | 2.15 | 0.143 |
| Cit | 0.018 | 0.11 | 0.03 | 0.872 | 0.139 | 0.11 | 1.73 | 0.189 |
| Orn | -0.066 | 0.03 | 3.87 | **0.049***  | -0.074 | 0.03 | 5.42 | **0.020***  |
| Pro | -0.027 | 0.01 | 3.64 | 0.057 | -0.016 | 0.01 | 1.64 | 0.201 |

Data with statistical difference is shown (bold*) obtained by ordinal logistic regression fit for BMI or WHtR. β: beta coefficient, S. E.: standard error, χ2: chi square, Glucose: Gluc, Ins: insulin, HOMA-IR: homeostasis model assessment insulin resistance, TAG: triglycerides, TC: total cholesterol, LDL-c: low-density lipoprotein cholesterol, HDL-c: high density lipoprotein cholesterol, C0: carnitine, C2: acetyl carnitine, C4: isobutyril carnitine, C6: hexanoyl carnitine, C8: octanoyl carnitine, C10: decanoyl carnitine, C12: dodecanoyl carnitine, C14: tetradecanoyl carnitine, C16: palmitoyl carnitine, C18: octadecanoylarnitine (stearoylcarnitine), C3: propionyl carnitine, C5: isovaleryl carnitine, Gly: glycine, Ala: alanine, Met: methyonine, Leu: leucine, Val: Valine, Phe: phenylalanine, Tyr: tyrosine, Arg: arginine, Cit: citrulline, Orn: ornitine, Pro: proline.

Finally, different cytokines were measured to determine if subclinical immunological response could be installed on school-aged children with total or central obesity. Levels of inflammatory cytokines showed no major differences between groups in both classifications (Fig 2).

When we analyzed by bivariate analysis only MIP-1β was higher in CVR than NG (Table 5); but after multiple adjustments this difference disappeared.

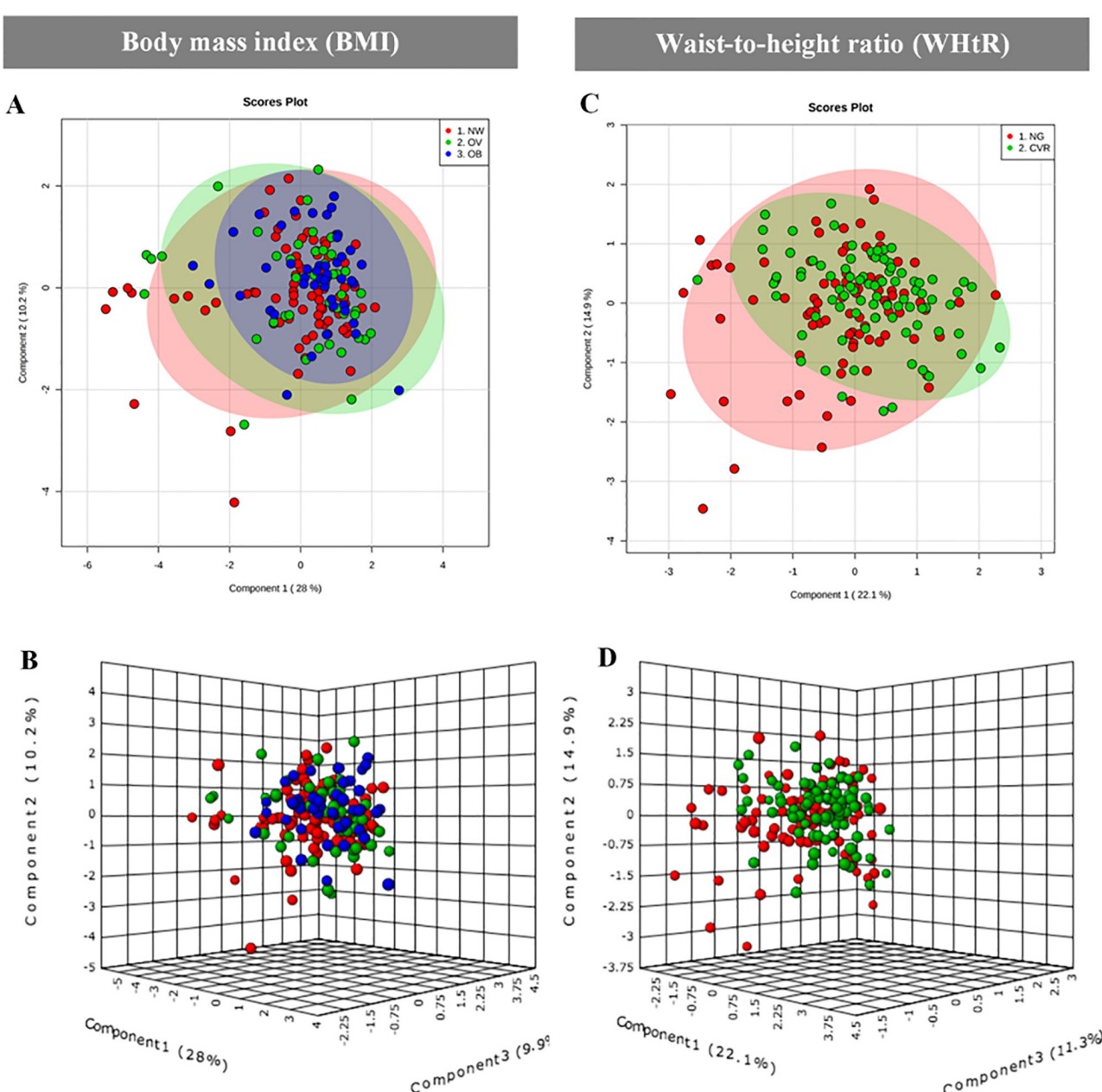

**Fig 2. Cytokines serum profile in schoolchildren classified by BMI or WHtR.** (A- B) BMI classification; (C-D) WHtR classification. PLS-DA plots show the maximum covariance between cytokines concentrations in children groups.

## Discussion

Rationale of this study was to simultaneously compare circulating metabolomic and inflammatory biomarkers in a cross-sectional study in healthy school-aged children classified by ranks of either BMI or WHtR. Our results support that metabolic disturbances precede changes in the inflammatory biomarkers in children with increased adiposity. It is well established that increased adiposity impacts metabolic homeostasis [22], and some of the involved lipid metabolites induce inflammatory response in the adipose tissue and other organs as obesity evolves.

**Table 5. Cytokines concentration in serum by BMI and WHtR.**

| Cytokines | BMI classification | | | | | | WHtR classification | | | |
|---|---|---|---|---|---|---|---|---|---|---|
| | NW | | OV | | OB | | NG | | CVR | |
| | Mean ± SD | CI 95% | Mean ± SD | CI 95% | Mean ± SD | CI 95% | Mean ± SD | CI 95% | Mean ± SD | CI 95% |
| IL-1α | 12 ± 21 | 7–16 | 13 ± 25 | 6–21 | 7 ± 9 | 5–10 | 11 ± 20 | 7–16 | 11 ± 19 | 7–15 |
| IL-1β | 2 ± 4 | 1–3 | 2 ± 2 | 1–2 | 1 ± 1 | 1–2 | 2 ± 4 | 1–3 | 1 ± 2 | 1–2 |
| IL-6 | 5 ± 11 | 3–8 | 6 ± 13 | 2–9 | 3 ± 6 | 1–5 | 5 ± 11 | 3–7 | 4 ± 10 | 2–7 |
| TNF α | 7 ± 6 | 6–8 | 7 ± 3 | 6–8 | 7 ± 3 | 6–8 | 7 ± 5 | 5–8 | 7 ± 3 | 6–8 |
| IL-17 | 3 ± 4 | 2–4 | 3 ± 2 | 2–3 | 3 ± 2 | 2–3 | 3 ± 4 | 2–4 | 3 ± 2 | 2–3 |
| IL-1RA | 18 ± 43 | 8–27 | 10 ± 15 | 5–14 | 5 ± 6 | 4–7 | 18 ± 41 | 9–27 | 7 ± 10 | 5–9 |
| IL-10 | 3 ± 9 | 1–5 | 2 ± 5 | 1–4 | 2 ± 5 | 1–3 | 3 ± 8 | 1–5 | 2 ± 5 | 1–3 |
| IL-12p40 | 9 ± 17 | 5–13 | 7 ± 9 | 4–10 | 5 ± 5 | 3–6 | 10 ± 17 | 6–14 | 5 ± 5 | 4–7 |
| IL-2 | 3 ± 6 | 2–4 | 3 ± 3 | 2–4 | 2 ± 2 | 1–2 | 3 ± 6 | 2–4 | 2 ± 2 | 1–2 |
| sIL-2RA | 32 ± 66 | 17–47 | 19 ± 36 | 8–29 | 34 ± 62 | 15–52 | 33 ± 64 | 19–47 | 25 ± 52 | 14–36 |
| VEGF | 105±123 | 78–133 | 91 ± 138 | 50–133 | 100 ± 113 | 67–134 | 97 ±118 | 71–123 | 103±129 | 75–131 |
| MIP-1 α | 6 ± 5 | 5–7 | 7 ± 5 | 5–8 | 7 ± 5 | 5–8 | 6 ± 56 | 5–8 | 7 ± 4 | 6–8 |
| MIP-1 β | 50 ± 19 | 46–54 | 54 ± 19 | 49–60 | 56 ± 23 | 49–63 | 51 ± 21 | 46–55 | **55 ± 20***| 51–60 |
| IP-10 | 342±186 | 300–385 | 375±192 | 317–433 | 383 ± 258 | 307–459 | 343±181 | 304–383 | 380 ± 233 | 330–431 |

Kruskal Wallis with Dunnett's multiple comparison tests were used to evaluate the parameters classified by BMI and, the U Mann Whitney test were used to evaluate parameters classified by WHtR. Single significant difference is signaled in bold* (p < 0.05).

The specific contribution of these alterations in the natural history of obesity comorbidities is still not well identified.

In this study, we analyzed and compared metabolomic and inflammatory biomarkers in children stratified by BMI as a proxy of total body adiposity or WHtR as a proxy of upper body fat distribution around the abdomen. We decided to compare results using both classifications in order to address actual controversy about the accuracy of these anthropometric measures to predict metabolic health status [23–26]. In our study we did not find significative differences using either BMI or WHtR for identification of children with either metabolic or inflammatory biomarkers disturbances.

Children with higher adiposity, either by BMI, showed increased glycemia in association to disturbances in lipid and amino acid metabolism, some of these results have been reported previously as a metabolic signature in obese children and adolescents [27–30].

Lipid metabolism in children with increased adiposity is characterized by higher concentrations of circulating triglycerides, L-palmitoylcarnitine, and free carnitine, revealing increased availability of fatty acids for oxidation. No evidence of beta-oxidation disfunction was present in these children since no accumulation of circulating short or medium-chain acylcarnitines species was found, making sense that fatty acids oxidation is a major source of energy [31,32] and consequently they are displacing the use of glucose by tissues, manifested as higher glycemia values [33].

Amino acid metabolism showed also significative changes in children with higher adiposity; glycine decreased, and valine, phenylalanine, and ornithine increased. These metabolic features have been identified previously [27,29]. Reduced circulating concentration of glycine may be explained by funneling glycine to gluconeogenesis induced by higher concentrations of fatty acids, displacing the incorporation of pyruvate to Krebs cycle [34], which may contribute to increased concentration of blood glucose. No clear explanation exists for high concentration of branched-chain amino acids (BCAA) such as valine that was found increased in children with obesity, confirming previous reports [35–39].

In comparison with metabolic disturbances, we did not find major differences in circulating inflammatory cytokines associated to increased adiposity in children classified either by BMI or WHtR. These findings contrast to a current report from Mărginean and *cols* where they noted an low-grade inflammatory status related to elevated counts of leukocytes, lymphocytes, and platelets in children with overweight and obesity [40]. However, they used a bivariate statistical analysis and no correction by age or sex of children was attempted. It is possible that younger children, as in our study, do not show the same cellular response. Inflammation is a complex tissular response that is frequently misinterpreted in obesity studies and further effort to homogenize terms such as "low-grade inflammation" must be accomplished in the literature [41]. Also, they did not find changes on glycemia levels indicating that adiposity in children may respond differentially according to lifestyles and genetic background. A cohort design is required to evaluate time course of inflammatory cytokine release to systemic circulation which have been proposed as independent cardiometabolic disease risk factors [42–44].

MIP-1β was the only significative inflammatory biomarker increased in children with CVR classification. MIP-1β is a chemokine which concentration has been correlated with waist circumference in young adults [45]. MIP-1β production is induced by palmitate in adipose tissue suggesting that this could be part of the mechanisms linking metabolism and inflammation [46].

Prevalence of alterations in blood pressure in children are still poorly studied and the role of these alterations in cardiovascular pathologies during later phases of life is still not known. It is possible that many of these early metabolic and functional differences in children may have a developmental programing origin during pregnancy and early childhood.

Our study has some limitations such as a relatively small sample size and the lack of information about lifestyles outside of the school, as well as dietary and exercise habits. Although we considered in the study the age and gender as changing factors of circulating metabolic mediators, further longitudinal studies in children cohorts must be done to clarify the trajectories of metabolic substrates and adipose tissue accumulation correlated with their lifestyles. Genetic factors must be also considered in metabolic trajectories analyses during childhood.

## Conclusion

Our findings suggest that school-aged children classified with abnormal adiposity show metabolic disturbances characterized by differential concentrations of circulating lipids and amino acids, that are not accompanied by systemic inflammatory response.

## Supporting information

**S1 Table. Metabolic profile in schoolchildren classified by BMI or WHtR.** Kruskal Wallis with Dunnett's multiple comparison test were used to evaluate the parameters classified by BMI and, the U Mann Whitney test were used to evaluate parameters classified by WHtR. Significant differences ($p < 0.05$) between groups in both types of classification are shown with different letters. Gluc: Glucose, Ins: insulin, HOMA-IR: homeostasis model assessment insulin resistance, TG: triglycerides, TC: total cholesterol, LDL-C: low-density lipoprotein cholesterol, HDL-C: high density lipoprotein cholesterol, C0: carnitine, C2: acetyl carnitine, C4: isobutyril carnitine, C6: hexanoyl carnitine, C8: octanoyl carnitine, C10: decanoyl carnitine, C12: dodecanoyl carnitine, C14: tetradecanoyl carnitine, C16: palmitoyl carnitine, C18: octadecanoylarnitine (stearoylcarnitine), C3: propionyl carnitine, C5: isovaleryl carnitine, Gly: glycine, Ala: alanine, Met: methyonine, Leu: leucine, Val: Valine, Phe: phenylalanine, Tyr: tyrosine, Arg: arginine, Cit: citrulline, Orn: ornithine, Pro: proline.
(DOCX)

## Acknowledgments

We thank to all participants in this project including: Nohemí Morán Díaz, Yunuén Pruneda Padilla, Marcial López Cervantes, Mercedes Gutiérrez Mares, Rocío Urbina Arronte, Samantha Escudero Gontes, Roberto Enrique Estrada Barragán, Bárbara Noemí Pantaleón Torres, Irma Cristina Carbajal Castillo, Andrea Siles Miranda, Cynthia Denise Camacho Robles, Erick Álvarez Álvarez, Inti Pérez Casillas, Fernanda Martínez.

## Author Contributions

**Conceptualization:** Ernestina Polo-Oteyza, Felipe Vadillo-Ortega.

**Data curation:** Erika Chavira-Suárez, Cecilia Rosel-Pech.

**Formal analysis:** Erika Chavira-Suárez, Cecilia Rosel-Pech, Isabel Ibarra-González, Marcela Vela-Amieva, Noemi Meraz-Cruz, Carlos Aguilar-Salinas, Felipe Vadillo-Ortega.

**Funding acquisition:** Felipe Vadillo-Ortega.

**Investigation:** Erika Chavira-Suárez, Ernestina Polo-Oteyza, Mónica Ancira-Moreno, Felipe Vadillo-Ortega.

**Methodology:** Cecilia Rosel-Pech, Mónica Ancira-Moreno, Isabel Ibarra-González, Marcela Vela-Amieva, Noemi Meraz-Cruz, Carlos Aguilar-Salinas, Felipe Vadillo-Ortega.

**Project administration:** Ernestina Polo-Oteyza, Felipe Vadillo-Ortega.

**Supervision:** Ernestina Polo-Oteyza, Felipe Vadillo-Ortega.

**Writing – original draft:** Erika Chavira-Suárez, Felipe Vadillo-Ortega.

**Writing – review & editing:** Erika Chavira-Suárez, Cecilia Rosel-Pech, Ernestina Polo-Oteyza, Mónica Ancira-Moreno, Isabel Ibarra-González, Marcela Vela-Amieva, Noemi Meraz-Cruz, Carlos Aguilar-Salinas, Felipe Vadillo-Ortega.

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
