## [Decision Letter · Decision Letter 0]

16 Apr 2020

PONE-D-20-01403

Simultaneous evaluation of metabolomic and inflammatory biomarkers in children with different body mass index (BMI) and waist-to-height ratio (WHtR)

PLOS ONE

Dear  Aithors 

Thank you for submitting your manuscript to PLOS ONE. After careful consideration, we feel that it has merit but does not fully meet PLOS ONE’s publication criteria as it currently stands. Therefore, we invite you to submit a revised version of the manuscript that addresses the points raised during the review proco and Interprate the statistical analysis as suggested by the Reviewer. Likewise please do the changes in the 

We would appreciate receiving your revised manuscript  When you are ready to submit your revision, Kindly resubmit with al the quereiess recified 

log on to https://www.editorialmanager.com/pone/ and select the 'Submissions Needing Revision' folder to locate your manuscript file.

To enhance the reproducibility of your results, we recommend that if applicable you deposit your laboratory protocols in protocols.io, where a protocol can be assigned its own identifier (DOI) such that it can be cited independently in the future. For instructions see: http://journals.plos.org/plosone/s/submission-guidelines#loc-laboratory-protocols

We look forward to receiving your revised manuscript.

Kind regards,

Nayanatara Arun Kumar

Academic Editor

PLOS ONE

Journal Requirements:

1. Thank you for including your competing interests statement; "

I have read the journal's policy and the authors of this manuscript have the following competing interests: EPO is the Director of Fondo Nestlé para la Nutrición in the Fundación Mexicana para la Salud.

The other authors have indicated they have no financial relationships relevant to this article to disclose."

We note that one or more of the authors are employed by a commercial company: Fondo Nestlé para la Nutrición in the Fundación Mexicana para la Salud.

Reviewers' comments:

Reviewer's Responses to Questions

**Comments to the Author**

1. Is the manuscript technically sound, and do the data support the conclusions?

Reviewer #1: Partly

Reviewer #2: Yes

2. Has the statistical analysis been performed appropriately and rigorously? 

Reviewer #1: Yes

Reviewer #2: Yes

3. Have the authors made all data underlying the findings in their manuscript fully available?

Reviewer #1: Yes

Reviewer #2: Yes

4. Is the manuscript presented in an intelligible fashion and written in standard English?

Reviewer #1: Yes

Reviewer #2: Yes

5. Review Comments to the Author

Reviewer #1: Vadillo-Ortega and collaborators have studied changes in metabolic and inflammatory markers in obese children. Since these children show altered some of the metabolic markers but not those of inflammation, they conclude that the metabolic changes must precede the inflammatory ones.

It is difficult to follow the changes that occur, since the p values obtained in the ANOVAS or in the tests used are not indicated. These p-values must be included in order to be able to properly assess the changes that have occurred. In addition, although the differences obtained in the post-hoc tests are indicated, it is not clear what they refer to specifically. Perhaps it would be better to indicate the differences in the post-hoc tests, and between which groups, at the bottom of the tables.

Differences with reference 40 should be commented on more extensively, since the parameters evaluated were different. It would be interesting to evaluate leukocyte, lymphocyte, erythrocyte, platelet, CRP, and transaminase levels to see if the differences obtained in Marginean's and collaborators' study are also replicated in the children of the present study. I find it difficult, although this is a personal opinion, to think that only metabolic changes are the inducers of obesity in these children.

Minor points:

All numbers in all tables should always include a maximum of three significant digits. For example, in Table 1, the systolic pressure values include a figure of 104±10.5 and a figure of 108.2±12. The data should be more uniform, e.g. 104±11 and 108±12.

In table 1 the % is expressed and then the amount in parentheses; the amount should be indicated first and the percentage in parentheses (and the heading should be N (%)).

In Table 4, since the insulin values have been measured, they should be included.

Why were only 12 amino acids measured?

There is no difference between groups when comparing the different acyl-carnitines. It might be better to include this table as a supplementary table.

Reviewer #2: Manuscript describes an interesting cross sectional study that measured metabolic and inflammatory biomarkers in Mexican children with different levels of adiposity. Targeted metabolomics and appropriate statistical tools have been used for data analysis. Even with acknowledged limitations of relatively small sample size, lack of life-style and dietary data, study provides good information which would be very useful for future studies.

6. PLOS authors have the option to publish the peer review history of their article (what does this mean?). If published, this will include your full peer review and any attached files.

Reviewer #1: No

Reviewer #2: No

---

## [Author Response · Author response to Decision Letter 0]

30 May 2020

Responses to Reviewer #1: 

Corrections were made and highlighted in green, as they appear in the file named “Revised manuscript with track changes”. 

1. p-values with statistical significance were included in tables. We modified all Tables to clarify their content. Old table 4 is now Supplementary Table 1 (S1). New Table 4 describes multivariate analysis.

2. Two paragraphs were added in Discussion addressing Marginean's study (Lines 299-304).

3. All numbers in tables were converted to include a maximum of three significant digits. 

4. Insulin values were added in New Table 4.

5. Number of measured amino acids was limited by the pre-designed commercial kit. However, we selected this kit because all groups of amino acids were represented according to their metabolic fate.

6. Bivariate analysis that included acylcarnitines concentrations (old Figure 4) was included in Supplementary Table 1

---

## [Editor Report · Decision Letter 1]

16 Jun 2020

PONE-D-20-01403R1

Simultaneous evaluation of metabolomic and inflammatory biomarkers in children with different body mass index (BMI) and waist-to-height ratio (WHtR)

PLOS ONE

Dear Dr. Felipe Vadillo-Ortega

Thank you for submitting your manuscript to PLOS ONE. After careful consideration, we feel that it has merit but does not fully meet PLOS ONE’s publication criteria as it currently stands. Therefore, we invite you to submit a revised version of the manuscript that addresses the points raised during the review process.

   Dear authors 

           Based on the reviewers comment on this manuscript. considerin. Kindly submit the revised at the earlies

Please submit your revised manuscript by 30th June If you will need more time than this to complete your revisions, please reply to this message or contact the journal office at plosone@plos.org. When you're ready to submit your revision, log on to https://www.editorialmanager.com/pone/ and select the 'Submissions Needing Revision' folder to locate your manuscript 

We look forward to receiving your revised manuscript.

Kind regards,

Nayanatara Arun Kumar

Academic Editor

PLOS ONE

---

## [Author Response · Author response to Decision Letter 1]

17 Jul 2020

Mexico City, May 14,2020

Rebuttal letter

Responses to Reviewer #1: 

Corrections were made and highlighted in green, as they appear in the file named “Revised manuscript with track changes”. 

1. p-values with statistical significance were included in tables. We modified all Tables to clarify their content. Old table 4 is now Supplementary Table 1 (S1). New Table 4 describes multivariate analysis.

2. Two paragraphs were added in Discussion addressing Marginean's study (Lines 299-304).

3. All numbers in tables were converted to include a maximum of three significant digits. 

4. Insulin values were added in New Table 4.

5. Number of measured amino acids was limited by the pre-designed commercial kit. However, we selected this kit because all groups of amino acids were represented according to their metabolic fate.

6. Bivariate analysis that included acylcarnitines concentrations (old Figure 4) was included in Supplementary Table 1 

Additional changes:

Other corrections were made, they appear turquoise highlighted in the file named “Revised manuscript with track changes”.

1. Line 4: biomarkers instead of markers.

2. Line 8: corresponding author affiliation number.

3. Line 139: Gluc and TAG initials instead of Glucose and TG.

4. Line 149: initials of Insulin (Ins) was added.

5. Lines 151 and 152: initials of Gluc and Ins instead of the complete name.

6. Line 181, 192, 214, and 246: schoolchildren instead of school children. 

7. Line 185, 196: initial definitions were included in the figure legend.

8. Line 189 -190, 250: NG initials instead of N.

9. Line 219- 227: all paragraph was modified.

10. Line 289 and 294: tyrosine and Val words were changed for phenylalanine, and ornithine such as valine (complete word).

---

## [Editor Report · Decision Letter 2]

6 Aug 2020

Simultaneous evaluation of metabolomic and inflammatory biomarkers in children with different body mass index (BMI) and waist-to-height ratio (WHtR)

PONE-D-20-01403R2

Dear Dr. Vadillo-Ortega,

We’re pleased to inform you that your manuscript has been judged scientifically suitable for publication and will be formally accepted for publication once it meets all outstanding technical requirements.

Kind regards,

Nayanatara Arun Kumar

Academic Editor

PLOS ONE

Additional Editor Comments (optional):

Dear authors

Very sorry for the delay in the decision from my side. The reviewers comments have been addressed and i congragulate all the auhtors . this paper can be accepted for the publication .

with best wishes

Dr. Nayana tara Arun KUmar

Associate Professor in Physiology

Kasturba Medical College, Mangalore
---

## [Editor Report · Acceptance letter]

12 Aug 2020

PONE-D-20-01403R2 

Simultaneous evaluation of metabolomic and inflammatory biomarkers in children with different body mass index (BMI) and waist-to-height ratio (WHtR) 

Dear Dr. Vadillo-Ortega:

I'm pleased to inform you that your manuscript has been deemed suitable for publication in PLOS ONE. Congratulations! Your manuscript is now with our production department. 

Kind regards, 

on behalf of

Dr. Nayanatara Arun Kumar 

Academic Editor

PLOS ONE